# Effects of Vitamin D on Satellite Cells: A Systematic Review of In Vivo Studies

**DOI:** 10.3390/nu14214558

**Published:** 2022-10-29

**Authors:** Muhammad Subhan Alfaqih, Vita Murniati Tarawan, Nova Sylviana, Hanna Goenawan, Ronny Lesmana, Susianti Susianti

**Affiliations:** 1Biomedical Science Master Program, Faculty of Medicine, Universitas Padjadjaran, Jl. Prof Eyckman No.38, Bandung 45363, Indonesia; 2Department of Biomedical Sciences, Faculty of Medicine, Universitas Padjadjaran, Jatinangor 45363, Indonesia; 3Central Laboratory, Universitas Padjadjaran, Jatinangor 45363, Indonesia

**Keywords:** vitamin D, satellite cells, skeletal muscle, in vivo

## Abstract

The non-classical role of vitamin D has been investigated in recent decades. One of which is related to its role in skeletal muscle. Satellite cells are skeletal muscle stem cells that play a pivotal role in skeletal muscle growth and regeneration. This systematic review aims to investigate the effect of vitamin D on satellite cells. A systematic search was performed in Scopus, MEDLINE, and Google Scholar. In vivo studies assessing the effect of vitamin D on satellite cells, published in English in the last ten years were included. Thirteen in vivo studies were analyzed in this review. Vitamin D increases the proliferation of satellite cells in the early life period. In acute muscle injury, vitamin D deficiency reduces satellite cells differentiation. However, administering high doses of vitamin D impairs skeletal muscle regeneration. Vitamin D may maintain satellite cell quiescence and prevent spontaneous differentiation in aging. Supplementation of vitamin D ameliorates decreased satellite cells’ function in chronic disease. Overall, evidence suggests that vitamin D affects satellite cells’ function in maintaining skeletal muscle homeostasis. Further research is needed to determine the most appropriate dose of vitamin D supplementation in a specific condition for the optimum satellite cells’ function.

## 1. Introduction

Vitamin D is a prohormone that has two main inactive isoforms, namely vitamin D2 (ergocalciferol) and vitamin D3 (cholecalciferol) [1,2]. Vitamin D2 is obtained from ultraviolet (UV) irradiation of ergosterol, a steroid that is found in some plants and fungi. Meanwhile, vitamin D3 is obtained mainly from UV irradiation of 7-dehydrocholesterol in the skin [3,4]. In addition, vitamin D3 is also obtained to a small extent from dietary intake such as oily fish, meat, or egg [2,3].

Vitamin D either obtained from UV exposure or food is then hydroxylated in the liver by vitamin D-25-hydroxylase to become 25-hydroxyvitamin D (25(OH)D) or also called calcidiol. 25(OH)D is a stable metabolite in the blood and best reflects exposure and absorption of Vitamin D. Therefore, 25(OH)D is used as an indicator of vitamin D status [1,2]. Furthermore, 25(OH)D needs to be converted by the enzyme 25(OH)D-1-OHase (CYP27B1) in the kidneys to become 1.25-dihydroxyvitamin D (1.25(OH)_2_D), the active form of vitamin D or also called calcitriol [5].

The classical function of vitamin D that has long been recognized is its role in the regulation of calcium and phosphate homeostasis and bone metabolism [6,7]. In the last few decades, some studies have indicated the non-classical function of vitamin D in various organs including skeletal muscle [8,9].

Satellite cells are skeletal muscle stem cells located between the basal lamina and sarcolemma which play a pivotal role in skeletal muscle growth and regeneration [10,11]. In homeostatic condition, satellite cells typically are in a quiescent state. The satellite cells will be activated when there are stimuli for skeletal muscle regeneration or hypertrophy, such as during an injury or exercise. Activated satellite cells will become myoblasts which proliferate and differentiate further to form new muscle fibers or fuse with preexisting muscle fibers [11,12]. This process is regulated by several myogenic regulatory factors (such as Myf5, MyoD, MRF4, and myogenin) [13,14]. In addition, some activated satellite cells can also undergo self-renewal and return to a quiescent state to replenish the satellite cell population (Figure 1) [15].

Several studies have shown that satellite cells express vitamin D receptor (VDR) [16,17,18]. Therefore, vitamin D may play a role in regulating satellite cells’ activity and function. A previous systematic review has discussed the effects of active vitamin D on myogenesis in vitro [19]. However, in vitro models cannot fully mimic the microenvironment of in vivo models [20,21]. Satellite cells’ fate is strongly influenced by their niche. Signals and properties of muscle fibers, basal lamina, as well as microvasculature, and surrounding interstitial cells influence the regulation of satellite cells’ function [22]. When satellite cells are removed from their in vivo niche and cultured in vitro, the satellite cells are activated and committed to proliferation and differentiation, thereby losing their stem cell properties [23]. This systematic review aims to collect the latest evidence from in vivo studies regarding the role of vitamin D on the satellite cells.

## 2. Methods

### 2.1. Search Strategy and Selection Process

This systematic review was constructed according to the Preferred Reporting Items for Systematic Reviews Meta-Analysis (PRISMA) guidelines. Relevant articles were identified through the online databases on Scopus, MEDLINE, and Google Scholar using the keywords (“vitamin D” OR “cholecalciferol” OR “calcitriol”) AND (“satellite cell*” OR “skeletal muscle stem cell*”) AND (“in vivo”). The search for articles was also performed manually to ensure getting as many relevant studies as possible. The article search was conducted in July 2022 and repeated in September 2022. The following inclusion and exclusion criteria were used in selecting articles:

Inclusion criteria

Articles are written in EnglishPublished in the last ten yearsIntervention with vitamin D supplementation or vitamin D deficiencyAssess the number or function of satellite cells

Exclusion criteria

Review articlesIn vitro studiesVitamin D supplementation in combination with other interventions (drugs/nutrients/exercise)The study does not state the markers of the satellite cell’s function

The measured outcome of this review is the number of satellite cells characterized by Pax7+ cell quantification or the marker that regulates satellite cells’ activity and function.

### 2.2. Data Extraction

Data from the included articles were extracted using a standardized form that had been determined and approved by all authors. Any questions and discrepancies were resolved by discussion among the authors. Data extracted included: title, author, year of publication, animal’s species, animal’s age, number of animals per group, route of administration and dosing of vitamin D, and major findings related to satellite cells.

### 2.3. Risk of Bias Assessment

The risk of bias was evaluated according to the Systematic Review Centre for Laboratory Animal Experimentation (SYRCLE’s) risk of bias tool for animal studies [24,25]. 

## 3. Results

### 3.1. Study Selection

The initial search from Scopus, MEDLINE, and Google Scholar obtained 1421 records. A manual search of the bibliography of the relevant articles yielded four eligible articles. After reducing duplication and screening using titles and abstracts, 48 full-text articles were analyzed further. Thirty-five articles were excluded and a total of thirteen articles were used in this systematic review. The study selection flow chart is shown in Figure 2.

### 3.2. Study Characteristics

From thirteen in vivo studies included, six studies used rats, four studies used mice, two studies used pigs, and one study used chickens as experimental animals. The effect of vitamin D supplementation or restriction was assessed in growing animals, animals with muscle injuries, and aging or diseased animals (Table 1).

### 3.3. Risk of Bias Assessment

Table 2 shows the summary of the risk of bias assessment. Seven of thirteen studies described random treatment in the allocation of the subject group [27,28,30,31,33,35,36]. No study has declared a blinding procedure in the intervention and outcome assessment. In the included in vivo studies, several parameters of the risk of bias assessment were not reported. In line with these findings, a previous study reported that the prevalence of reporting the risk of bias in animal studies is still low [40].

### 3.4. Effect of Vitamin D on Satellite Cells during Prenatal Development and Postnatal Growth 

Studies on pigs showed that maternal vitamin D supplementation during pregnancy and lactation improves vitamin D status in newborn and weaning offspring [28,30]. Improved vitamin D status in the pig’s offspring is associated with an increase in the number of primary muscle fibers at birth. However, the number of satellite cells in each muscle fiber was not different from the control [30]. Vitamin D may play a role in muscle growth and development in the early life period not by increasing the number of satellite cells but by increasing the functional capacity of satellite cells to form new muscle fibers. Zhou et al. reported that maternal vitamin D supplementation increased the cross-sectional area of muscle fibers in weaning piglets. In addition, there was also an increase in the MyoD and myogenin mRNA expression, markers of the satellite cell differentiation [28].

A recent study by Reis et al. showed that maternal vitamin D deficiency in rats may affect satellite cell count and skeletal muscle morphology in adult offspring. In that study, maternal vitamin D deficiency selectively increased the number of satellite cells and myogenic regeneration factor in type 2 muscle fibers in adult male offspring. This finding may be due to an increase in local vitamin D production in adult age to compensate for the limited intrauterine vitamin D [31].

Hutton et al. demonstrated that dietary vitamin D supplementation in newly hatched chicks increased mitotically active satellite cells (Pax7+, BrdU+). In addition, the density of Pax7 + satellite cells in fast twitch muscle also tended to increase, but it was not statistically significant (*p* = 0.07) [27].

In young adult rats, where muscle growth is still occurring but at a slower rate, vitamin D deficiency is associated with decreased MyoD mRNA expression [29].

### 3.5. Effect of Vitamin D on Satellite Cell during Muscle Injury

Yu et al. used cyp27b1 knockout mice to determine the effect of 1.25(OH)_2_D deficiency on skeletal muscle regeneration. In the injured muscle of cyp27b1 knockout mice, there was a significant decrease in the number of centrally nucleated fibers and the mRNA expression of Myf5, MyoD, and MyHC [34]. Newly regenerated myofibers are characterized by their centrally located nucleus [41,42]. In that study, decreased centrally nucleated myofiber in cyp27b1 knockout mice indicated that 1.25(OH)_2_D deficiency was associated with decreased skeletal muscle regeneration. The early stages of skeletal muscle regeneration involve the activation of satellite cells. MyoD and Myf5 are myogenic factors that play a role in the activation of satellite cells [43,44,45]. Thus, decreased skeletal muscle regeneration in 1.25(OH)_2_D-deficient mice may partly be due to impairment of satellite cell activation.

On the other hand, Srikuea et al. reported that intramuscular injection of 1.25(OH)_2_D at a supraphysiological dose (1 g/kg mouse body weight) in the injured muscle resulted in decreased satellite cell differentiation and inhibition of skeletal muscle regeneration [33]. Moreover, Stratos et al. showed that, in mice with injured muscle, subcutaneous administration of high doses of vitamin D3 (332,000 IU/kg body weight) had no effect on the number of satellite cells [32]. 

### 3.6. Effect of Vitamin D on Satellite Cells during Aging and Diseased States

Faria et al. assessed the effects of vitamin D deficiency in naturally aging rats (24 months old ......rats) [35]. In aging rats with vitamin D deficiency, there was a decrease in the protein expression of the Nocth1 transmembrane fragment and the mRNA expression of Hes1, the target gene of Notch signaling [35]. Notch signaling is one of the key regulators of stem cells in various tissues, including skeletal muscle. In intact skeletal muscle, Notch signaling plays a role in maintaining the quiescent state of skeletal muscle stem cells (satellite cells). Meanwhile, in regenerating muscle, Notch signaling plays a role in self-renewal and expansion of the satellite cell population [46,47]. Vitamin D deficiency in aging rats was also associated with decreased mRNA expression of bone morphogenetic protein 4 (Bmp4) and fibroblast growth factor-2 (Fgf-2), markers that play a role in satellite cell proliferation [35].

Tamura et al. showed that in diabetic mice, vitamin D deficiency augments decreased Pax7 mRNA expression [37]. Pax7 is a transcription factor that plays a critical role in satellite cells’ function. The absence of Pax7 results in the inability of satellite cells to undergo proliferation and self-renewal and lead to a decrease in the population of satellite cells due to precocious differentiation [48]. Pax7 also plays a role in the specification of satellite cells. Deletion of Pax7 also causes alternative stem cell development and leads to hematopoietic or brown adipogenic commitment [49,50].

Cheung et al. used a mouse model of infantile nephropathic cystinosis (Ctsn-/-) which had vitamin D deficiency. In Ctsn-/- mice, there was muscle atrophy and increased adipose tissue browning. Vitamin D repletion increased Pax7, and MyoD mRNA expression and reduced muscle atrophy and adipose tissue browning in these mice [39]. 

The use of glucocorticoids can induce skeletal muscle atrophy. Kinoshita et al. examined the effect of eldecalcitol (activated vitamin D analog) on rats receiving glucocorticoids. The study showed that eldecalcitol administration for two weeks increased pax7 and myogenin mRNA expression [36].

## 4. Discussion

In the neonatal period, skeletal muscle undergoes a high growth rate that involves high protein synthesis accompanied by a rapid increase in myonuclei. Satellite cells contribute to the addition of myonuclei to growing muscle fibers in the postnatal period [51]. This process of skeletal muscle growth depends on the proliferation of satellite cells. In 4-day-old rats with a chronically unloaded hindlimb, there was impaired growth of the soleus muscle associated with a decrease in mitotically active satellite cells [52]. The rapid growth of skeletal muscle during early life requires adequate nutrition. Nutrition has a critical influence in providing components for muscle mass synthesis and various signaling involved in the muscle fiber anabolism [51]. 

Vitamin D is a micronutrient that plays a role in maintaining various body functions throughout life [53]. Srikuea et al. demonstrated that satellite cells are the target cells of vitamin D action and the response of satellite cells to vitamin D varies depending on age. There is a decrease in satellite cell response to vitamin D in aged muscles compared to muscles in the developmental age [33]. These findings support the importance of vitamin D signaling in early life when satellite cell activity is high. Improved vitamin D status is associated with increased proliferation and myogenic capacity of satellite cells in the early weeks of life [27,28]. However, studies do not support the effect of vitamin D signaling on satellite cell number during muscle growth in the early life period [27,30]. A possible explanation is that the high rate of satellite cell proliferation in the early life period plays a role in providing new myonuclei to growing muscles and not increasing the number of satellite cell reserves.

Acute injury involves sudden changes in the form of damage to muscle fibers, infiltration of inflammatory cells, edema, and damage to surrounding tissues. All of these lead to a change in the niche and trigger the activation of satellite cells [54]. Vitamin D signaling appears to influence satellite cells’ function in skeletal muscle regeneration. In mice with vitamin D deficiency, there was a decrease in markers of activation and differentiation of satellite cells [34]. However, high-dose vitamin D supplementation without considering baseline vitamin D status leads to impaired satellite cell differentiation, delayed muscle fiber formation, and fibrosis formation in regenerating muscle [33]. Dosing appears to be a crucial issue when administering vitamin D during muscle regeneration. In vitro studies suggest that the administration of vitamin D supports muscle regeneration in a dose-dependent manner. However, at very high doses, it inhibits muscle formation [55,56,57].

In aging, there are some changes in the satellite cell niche, which causes the satellite cells to lose their quiescence and tend to differentiate prematurely [58]. Exposure of satellite cells in aging mice with serum from young mice can restore the regenerative function of satellite cells [59]. Aging is associated with decreased Notch signaling, a master regulator in maintaining the quiescent state of satellite cells [58]. Decreased Notch signaling in aged rats with vitamin D deficiency suggests that vitamin D may play a role in the regulation of Notch signaling in aging [35]. In in vitro study, Olsson et al. showed that the administration of vitamin D to human-derived myoblasts increased Hes1 mRNA expression, the gene target of Notch signaling [16]. One possible mechanism for vitamin D to regulate Notch signaling is its role in increasing Forkhead box O3 (FOXO3) expression [16]. A previous study demonstrated that FOXO3 is expressed in quiescent satellite cells. FOXO3 modulates Notch signaling by directly increasing Notch receptor expression. The FOXO3-Notch axis is required for satellite cell self-renewal by restoring satellite cell quiescence in regenerating muscle [60].

Vitamin D deficiency in aged rats was also associated with decreased Bmp4 and Fgf2 mRNA expression [35]. BMP signaling plays a role in increasing the satellite cell pool by promoting satellite cell proliferation and preventing precocious differentiation [61]. Stantzou et al. showed that inhibition of BMP signaling decreases the satellite cell pool [62]. Meanwhile, Fgf2 enhances satellite cell proliferation without suppressing differentiation [63]. Faria et al. assumed that the aged rats in their study experienced discrete regeneration episodes due to daily damage, so satellite cell proliferation was needed [35]. In another study, Fgf2 expression increased in aging muscle and triggered satellite cell proliferation and myogenic differentiation in homeostatic conditions. This causes satellite cell depletion and reduced muscle regeneration capacity [64]. Further studies are needed to confirm the role of vitamin D supplementation on Fgf2 in aged rats.

In chronic illness, there may be a chronic injury to the muscle depending on the severity and type of disease. Alterations in energy metabolism, inflammation, or restriction of movement can be factors that cause changes in satellite cells’ activity in diseased states [65,66]. Han et al. reported that an increase in extracellular adenosine (eADO) in diabetic mice decreased the regenerative function of satellite cells [67]. Satellite cells cultured on a high-glucose medium showed decreased proliferation and expression of Pax7, MyoD, and myogenin proteins [68]. In diseased experimental animal models, vitamin D deficiency aggravates the impaired function of satellite cells. Meanwhile, vitamin D supplementation ameliorates the impaired function of these satellite cells. Thus, it is important to pay attention to vitamin D status in various chronic diseases. Further research is needed to determine the dose of vitamin D administration to improve satellite cell function in the setting of chronic disease.

## 5. Conclusions

In vivo studies support a direct role of vitamin D on satellite cells’ function during muscle growth, injury, aging, or chronic disease. Vitamin D appears to increase satellite cell proliferation in the early life period during rapid muscle growth. Adequate vitamin D status is required to support the satellite cells’ function in skeletal muscle regeneration during acute injury. However, the administration of high doses of vitamin D decreases satellite cell differentiation and delays new muscle fiber formation. Vitamin D deficiency in aging was associated with the decrease in Notch signaling resulting in satellite cells losing their quiescent and differentiating prematurely. Vitamin D supplementation ameliorates the impairment of satellite cell function in chronic disease. Thus, to provide optimal effects on satellite cells’ function, it is necessary to administer vitamin D at a dose according to the physiological needs of each individual. Further research is needed to determine the most appropriate dose and duration of vitamin D supplementation in the various age groups and specific conditions such as in early life, injury, aging, or chronic disease.

## Figures and Tables

**Figure 1 nutrients-14-04558-f001:**
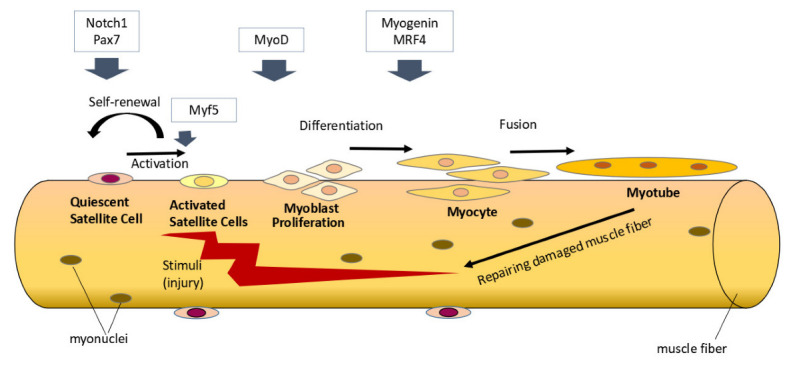
Satellite cells’ activity in skeletal muscle regeneration. Under homeostatic conditions, satellite cells are typically in a quiescent state and express Pax7. Notch signaling plays a role in maintaining the quiescent state of satellite cells. When there are stimuli for skeletal muscle regeneration, various myogenic regulatory factors (Myf5, MyoD, myogenin, and MRF4) regulate satellite cells’ activation, proliferation, and differentiation to form new muscle fibers. Some satellite cells will undergo self-renewal and return to a quiescent state to replenish the satellite cell population. Notch signaling through its regulation of Pax7 plays a role in promoting satellite cell self-renewal.

**Figure 2 nutrients-14-04558-f002:**
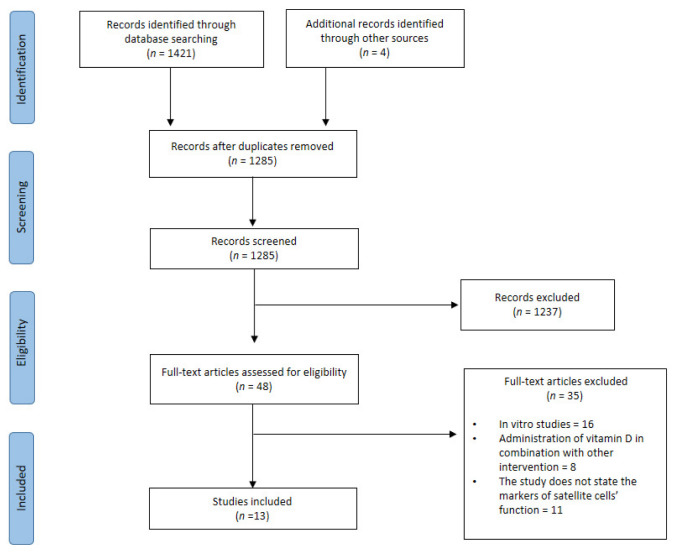
Study selection flow adapted from Preferred Reporting Items for Systematic Reviews and Meta-Analyses (PRISMA) guideline [26].

**Table 1 nutrients-14-04558-t001:** Characteristics of selected in vivo studies stratified based on the characteristics of experimental animal models.

References	Species	Vitamin D form, Dose, Duration	Importance
Animals in the period of prenatal development and postnatal growth			
Hutton et al., 2014 [27]	Male, Ross 708 broiler chicks (*n* total = 150)	2240 IU vitamin D3 per kg diet + 2760 IU of 25(OH)D_3_ per kg diet for 49 days	↑ mitotically active satellite cells (Pax7+; BrdU+), tended to ↑ density of Pax7+ satellite cells and Myf-5 + satellite cells in pectoralis mayor muscle
Zhou et al., 2016 [28]	White gilts (*n* = 10 per group)	50 µg/kg vitamin D3 + 50 µg/kg 25(OH)D_3_ given from mating to weaning	↑ MyoD1 and ↑ Myogenin mRNA expression in newborn and weaning piglets
Oku et al., 2016 [29]	Male Sprague Dawley rats (*n* = 6 per group)	Diet with vitamin D restriction for 28 days	↓ MyoD mRNA expression
Thayer et al., 2019 [30]	Sows and their progeny (*n* total = 69)	1500 IU/kg vitamin D3; 500 IU/kg vitamin D3 + 25 μg/kg 25(OH)D_3_; or 1500 IU/kg vitamin D3 + 50 μg/kg 25(OH)D_3_, during gestation and lactation	No difference in satellite cells number per fiber in pig’s muscle at birth
Reis et al., 2022 [31]	Male and female Wistar Hannover rats (*n* total = 20)	Maternal vitamin D deficiency	Following 180 days, only in Vitamin D deficiency male adult offspring, there are ↑ local calcitriol, ↑ CYP27B1, ↑ number of satellite cells, ↑ MyoD and ↑myogenin protein expression
		Animals with muscle injury	
Stratos et al., 2013 [32]	Male Wistar rats (*n* total = 56)	332,000 IU/kg body weight vitamin D3 single dose subcutaneous injection after muscle injury	↑ non myogenic cell proliferation, ↓ apoptosis; Not significant influence the number of satellite cells, ↑ extracellular matrix protein
Srikuea et al., 2016 [33]	Male C57BL/6 mice (*n* = 6 per group)	1.25(OH)_2_D_3_ 1 g/kg relative to tibialis anterior muscle wet weight (physiological dose) or 1.25(OH)_2_D_3_ 1 g/kg relative to mouse body weight (supraphysiological dose), intramuscular injection at day 4–7 post injury	↑ Vdr expresion in both doses; 1.25(OH)_2_D_3_ at a supraphysiological dose: ↓ satellite cell differentiation, delayed regenerative muscle fiber formation, and increased muscular fibrosis
Yu et al., 2021 [34]	Male Cyp27b1 knockout (KO) mice	Vitamin D deficiency in Cyp27b1 KO mice	↓ MyoD, ↓Myf5, ↓MyHC in tibialis anterior muscle of Cyp27b1 KO after injected with BaCl_2_
		Aging or diseased animal	
Faria et al., 2014 [35]	Male old Wistar rats (*n* = 10 per group)	AIN-93 M maintenance diet or a modified AIN-93 M diet with no vitamin D (to induce vitamin D deficiency) for 9 months	In vitamin D deficient group: ↓ Tibialis anterior weight, ↓ mRNA expression of marker of proliferation (Bmp4, Fgf-2, PCNA) ↓ Notch signaling activity
Kinoshita et al., 2015 [36]	Glucocorticoid-treated female Wistar rats (*n* = 5–9 per group)	Eldecalcitol (activated vitamin D3 analogue) for 2 or 4 weeks	↑ Pax7 and ↑ Myogenin mRNA expression during 2 weeks of treatment
Tamura et al., 2016 [37]	Diabetic C57BL/6 mice	Vitamin D-deficient diet for 6 weeks	↓ Pax7 mRNA expression
Nakaoka et al., 2019 [38]	Ovariectomized female Sprague–Dawley rats (*n* = 6 per group)	Vitamin D restriction for 28 days	↓ Myf-5 and ↓Myogenin mRNA expression
Cheung et al., 2020 [39]	C57BL/6 Ctns-/- mice (mouse model of infantile nephropathic cystinosis with Vitamin D insufficiency)	Supplementation with 25(OH)D + 1.25(OH)_2_D_3_ (75 μg/kg/ day + 60 ng/kg/day, respectively), for 6 weeks	Ameliorate the decreased gene expression of Pax7, and MyoD

VDR: Vitamin D Receptor; BrdU: Bromodeoxyuridine; Pax7: paired box 7; MyoD1: myogenic differentiation 1; Myf5: myogenic factor 5; MyHC: myosin heavy chain; PCNA: proliferating cell nuclear antigen; CYP27B1: Cytochrome P450 Family 27 Subfamily B Member 1; Bmp4: Bone morphogenetic protein 4; Fgf-1: Fibroblast growth factor-2; Igf-1: insulin-like growth factor-1; KO: knockout; mRNA: messenger RNA.

**Table 2 nutrients-14-04558-t002:** Risk of bias summary of in vivo studies.

References	Selection Bias	Performance Bias	Detection Bias	Attrition Bias	Reporting Bias	Other
1	2	3	4	5	6	7	8	9	10
Hutton et al. [27]										
Zhou et al. [28]										
Oku et al. [29]										
Thayer et al. [30]										
Reis et al. [31]										
Stratos et al. [32]										
Srikuea et al. [33]										
Yu et al. [34]										
Faria et al. [35]										
Kinoshita et al. [36]										
Tamura et al. [37]										
Nakaoka et al. [38]										
Cheung et al. [39]										
	 Low risk of bias	 No information/not applicable

Sequence generation (1), baseline characteristics (2), allocation concealment (3), random housing (4), intervention blinding (5), random outcome assessment (6), outcome blinding (7), incomplete outcome (8), selective outcome reporting (9) and other (10).

## Data Availability

The data presented in this study are available on request from the corresponding author.

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
