# Peer review of "Effects of Vitamin D on Satellite Cells: A Systematic Review of In Vivo Studies"

_nutrients, 2022, doi:10.3390/nu14214558_

Round 1

Reviewer 1 Report

The subject of the manuscript is interesting. Unfortunately, the description of the results and the discussion do not fully present the topic in question. First, selected in vitro studies concern C2C12 myoblasts or myoblasts obtained from satellite cells. The Authors of the study suggest in the title that they focuse on satellite cells. The Authors did not mention about the role of satellite cells are or myoblasts in the regeneration of skeletal muscles. Likewise, there is no introduction about vitamin D. The diagram shown is cell-general, not specific to satellite cells. The results are described very briefly. Information is not related to other. The Authors did not provide information on the research model as to whether the results relate to intact or regenerating muscles. The information is provided on a random basis and without an appropriate background. It is difficult to follow conclusions on the basis of the presented results. Therefore, I recommend rewriting the manuscript. 

Author Response

Response to Reviewer 1 Comments

Point 1: The subject of the manuscript is interesting. Unfortunately, the description of the results and the discussion do not fully present the topic in question. First, selected in vitro studies concern C2C12 myoblasts or myoblasts obtained from satellite cells. The Authors of the study suggest in the title that they focuse on satellite cells.

Response 1: Thank you very much for the valuable assessment. We agree with this comment. Therefore, we have expanded the scope of our study as stated in the revised title. In addition to focusing on satellite cells, we also focus on their function in skeletal muscle growth and regeneration, which involves the activity and function of the progeny of satellite cells. C2C12 cells were derived from satellite cells and are a known reliable model in studying the process of skeletal muscle regeneration, so studies using C2C12 are included in this review.

Point 2: The Authors did not mention about the role of satellite cells are or myoblasts in the regeneration of skeletal muscles. Likewise, there is no introduction about vitamin D

Response 2: Thank you for pointing this out. We have added this point according to the comment (Lines no. 31-55 and 56-65)

Point 3: The diagram shown is cell-general, not specific to satellite cells. 

Response 3: Thank you for pointing this out. We have revised the diagrams and the descriptions in figures 1 and 3

Point 4: The results are described very briefly. 

Response 4: We agree with the reviewer’s assessment. We have added a description regarding the results (Lines 128-320)

Point 5: Information is not related to other

Response 5: Thank you for pointing this out. We tried our best to improve the manuscript and made some changes in the manuscript. We appreciate for Reviewers’ warm work earnestly.

Point 6: The Authors did not provide information on the research model as to whether the results relate to intact or regenerating muscles

Response 6: Thank you very much for your valuable input. We have revised our manuscript to describe the function of satellite cells in skeletal muscle according to the research model. (Lines 236-240 and in Table 2)

Point 7: The information is provided on a random basis and without an appropriate background. It is difficult to follow conclusions on the basis of the presented results. Therefore, I recommend rewriting the manuscript. 

Response 7: Thank you very much for pointing this out. We tried our best to improve the manuscript and made some changes in the manuscript.

Reviewer 2 Report

In the article “Effects of Vitamin D on Satellite Cells: A Systematic Review”. It is a valuable review.  However, there are some shortcomings and questions.

1) The introduction should be revised. The Vitamin D vitamin signaling pathway and satellite

2) In the line 24-25: Further research is needed to determine the most appropriate dose of vita- min D supplementation in a specific age group for optimum satellite cells function. It could be revised as : Further research is needed to determine the most appropriate dose of vitamin D supplementation in a specific age group for the optimum satellite cells’ function.

Author Response

Response to Reviewer 2 Comments

Point 1: In the article “Effects of Vitamin D on Satellite Cells: A Systematic Review”. It is a valuable review.  However, there are some shortcomings and questions.

Response 1:.Thank you very much

Point 2:  The introduction should be revised. The Vitamin D vitamin signaling pathway and satellite

Response 2: Thank you for the suggestion. We have revised our introduction section according to the Reviewer's suggestion (Lines no. 30-71)

Point 3: In the line 24-25: Further research is needed to determine the most appropriate dose of vita- min D supplementation in a specific age group for optimum satellite cells function. It could be revised as : Further research is needed to determine the most appropriate dose of vitamin D supplementation in a specific age group for the optimum satellite cells’ function.

Response 3:  Thank you very much for pointing this out. We have revised this part according to the Reviewer’s comments (Lines 25-27 in the revised version)

Round 2

Reviewer 1 Report

The subject of the manuscript is interesting. Unfortunately, the description of the results and the discussion still do not fully present the topic in question.

Author Response

Response to Reviewer 1 Comments

Point 1: The subject of the manuscript is interesting. Unfortunately, the description of the results and the discussion still do not fully present the topic in question.

Response 1: Thank you very much for your valuable assessment. We agree with the reviewer. Therefore, we have revised the results and discussion to present the topic in question. We limit our discussion to in vivo studies, considering that the function of satellite cells is closely related to their niche, which is not fully represented in in vitro studies. We also include a graphical abstract to summarize the content and represent the article's topic.
